# Galectin-3 Mediates NETosis and Acts as an Autoantigen in Systemic Lupus Erythematosus-Associated Diffuse Alveolar Haemorrhage

**DOI:** 10.3390/ijms24119493

**Published:** 2023-05-30

**Authors:** Shih-Yao Chen, Chung-Teng Wang, Ching-Yi Chen, Pin-Yu Kuo, Chrong-Reen Wang, Ai-Li Shiau, Cheng-Hsi Chang, Chao-Liang Wu

**Affiliations:** 1Department of Nursing, College of Nursing, Chung Hwa University of Medical Technology, Tainan 71703, Taiwan; 06052@ms.hwai.edu.tw; 2Department of Microbiology and Immunology, College of Medicine, National Cheng Kung University, 1 University Road, Tainan 70101, Taiwan; 3Department of Internal Medicine, School of Medicine, College of Medicine, National Cheng Kung University, Tainan 70101, Taiwan; 4Department of Biochemistry and Molecular Biology, College of Medicine, National Cheng Kung University, Tainan 70101, Taiwan; 5Department of Internal Medicine, National Cheng Kung University Hospital, Tainan 70403, Taiwan; 6Ditmanson Medical Foundation Chia-Yi Christian Hospital, Zhongxiao Road 539, East District, Chiayi 60002, Taiwan; 7Department of Cardiovascular Surgery, Ditmanson Medical Foundation Chia-Yi Christian Hospital, Chiayi 60002, Taiwan

**Keywords:** systemic lupus erythematosus, NETosis, neutrophil extracellular traps, galectin-3, diffuse alveolar haemorrhage

## Abstract

Systemic lupus erythematosus (SLE) is a systemic autoimmune disease with enhanced NETosis and impaired degradation of neutrophil extracellular traps (NETs). Galectin-3 is a β-galactoside binding protein and is associated with neutrophil functions as well as involved in mediating autoimmune disorders. In this study, we plan to examine the associations of galectin-3 with the pathogenesis of SLE and NETosis. Galectin-3 expression levels were determined in peripheral blood mononuclear cells (PBMCs) of SLE patients for the association with lupus nephritis (LN) or correlation of SLE disease activity index 2000 (SLEDAI-2K). NETosis was observed in human normal and SLE and murine galectin-3 knockout (Gal-3 KO) neutrophils. Gal-3 KO and wild-type (WT) mice induced by pristane were used to evaluate disease signs, including diffuse alveolar haemorrhage (DAH), LN, proteinuria, anti-ribonucleoprotein (RNP) antibody, citrullinated histone 3 (CitH3) levels, and NETosis. Galectin-3 levels are higher in PBMCs of SLE patients compared with normal donors and positively correlated with LN or SLEDAI-2K. Gal-3 KO mice have higher percent survival and lower DAH, LN proteinuria, and anti-RNP antibody levels than WT mice induced by pristane. NETosis and citH3 levels are reduced in Gal-3 KO neutrophils. Furthermore, galectin-3 resides in NETs while human neutrophils undergo NETosis. Galectin-3-associated immune complex deposition can be observed in NETs from spontaneously NETotic cells of SLE patients. In this study, we provide clinical relevance of galectin-3 to the lupus phenotypes and the underlying mechanisms of galectin-3-mediated NETosis for developing novel therapeutic strategies targeting galectin-3 for SLE.

## 1. Introduction

Systemic lupus erythematosus (SLE) is a highly complex autoimmune disease with unknown etiology, possibly attributed to genetic and environmental factors and characterized by generation of circulating autoantibodies that target most organ systems including skin, joint, nerve, heart, lung, kidney, liver, intestinal tract, and hematological system [1]. Initial therapy for all patients is hydroxychloroquine, but during the disease flares, steroid use is needed, while the more severe manifestations with major organ involved require additional prescription of immunosuppressants such as cyclophosphamide [2]. Notably, high doses of steroids can cause adverse events such as infection, highlighting the needs for safe and effective targeted therapeutics for lupus patients such as the recently approved belimumab and anifrolumab. Although lupus nephritis (LN) therapy has been revolutionized by the advent of clinical trials with biologics such as rituximab (a chimeric anti-CD20 mAb) [3], ocrelizumab (a humanized anti-CD20 mAb) [4], and epratuzumab (an anti-CD22 mAb) [5], failed trials with frustrated results were demonstrated in such patients [3,4,5,6]. Indeed, further studies for elucidating the complex mechanisms can help develop novel therapeutics for SLE patients. 

NETosis, a programed neutrophil death, has been suggested for the pathogenesis of a variety of autoimmune and inflammatory diseases [7,8]. Although it is widely accepted that defective clearance of apoptotic or necrotic cells is the major source of autoantigens in SLE [9,10], impaired degradation of NETs and increased NET formation are found in these patients [11,12]. Although abnormalities in various immune cells have been clearly clarified in SLE, the role played by neutrophils remains to be elucidated [13].

Galectin-3 belongs to a group of lectins that specifically binds to β-galactoside carbohydrates with its carbohydrate-recognition domains (CRDs) and widely distributed in various cell types and tissues [14]. It has been found that the circulating levels of galectin-3 and anti-galectin-3 antibodies are higher in SLE patients than in healthy individuals, and correlated with LN [15,16] and lupus-specific skin lesions [17]. Galectin-3 can promote acute inflammation and act as an adhesion molecule for neutrophil extravasation during alveolar bacterial infection [18]. Pristane-injected galectin-3-deficient mice demonstrated diminished neutrophil numbers in the peritoneal cavities, indicating a critical role of galectin-3 in recruiting neutrophils to the inflammatory sites [19]. Furthermore, galectin-3 is also shown to participate in inflammation through modulating the functions of neutrophils, such as reactive oxygen species (ROS) production [20,21]. Through the aforementioned evidence, we suggest that galectin-3 governed by infiltrating neutrophils plays an important role in the pathogenesis of SLE.

Pristane-induced lupus is a well-established murine model that mimics human SLE [22]. Tetramethylpentadecane (2,6,10,14-tetramethylpentadecane, TMPD), commonly known as pristane, is a naturally occurring hydrocarbon oil that can induce chronic inflammatory responses when delivered once intra-peritoneally into the BALB/C or C57BL/6 strain [22]. After pristane injection, particular lupus manifestations such as arthritis can only be found in BALB/C mice, while diffuse alveolar haemorrhage (DAH) can be observed in C57BL/6 or C57BL/10 mice [22]. Despite not being a common morbidity like LN, DAH is a lethal presentation mostly associated with SLE patients, but rarely occurs in other autoimmune disorders [12]. The pristane-injected C57BL/6 or C57BL/10 mice not only provide a convenient research platform for studying the SLE disease but also serves as an alternative model for investigating the DAH manifestation.

Since a possible pathophysiological role of galectin-3 in SLE has not been fully clarified, we use galectin-3 knockout (Gal-3 KO) mice [23] induced by pristane, and sera and peripheral blood mononuclear cells (PBMCs) from SLE patients to study clinical associations and the underlying mechanisms. We demonstrated that galectin-3 was associated with NETosis, DAH, and LN in SLE. Levels of circulating anti-RNP antibody were lower in Gal-3 KO mice than their wild-type (WT) counterparts induced by pristane. Interestingly, we found the expression of citrullinated histone 3 (citH3) was decreased in lung tissue extracts from Gal-3 KO mice when compared with WT mice, and that galectin-3 residing in human neutrophils underwent NETosis. Spontaneous NETosis could only be observed in PBMCs of SLE patients in concert with immune complex deposition in NETs. We therefore proposed that galectin-3 is responsible for NETosis and plays an autoantigen role in SLE. 

## 2. Results

### 2.1. Increased Levels of Galectin-3 and Anti-Galectin-3 Antibody in PBMCs, Sera and Lung Tissues from SLE

The expression of galectin-3 was examined in PBMCs of patients with SLE. QRT-PCR revealed higher expression levels of galectin-3 in PBMCs from patients with SLE compared with those from normal individuals (Figure 1A). SLE patients with LN (LN-Yes) have higher expression of galectin-3 in comparison to those without LN (LN-Nil) (Figure 1A). A positive correlation between the levels of galectin-3 and SLEDAI-2K was also observed (Figure 1B), indicating higher galectin-3 expression correlates with more severe lupus activity. Despite no statistical significances (*p* = 0.0774), an increased trend of anti-galectin-3 antibody levels was observed in SLE patients’ sera compared with those in normal controls (Figure 1C). Since DAH is an early disease manifestation in pristane-induced lupus mice with the evident pathogenic mechanisms [22], we plan to determine whether galectin-3 is also involved in this distinct process. Figure 1D shows the expression levels of galectin-3 are higher in pristane-treated mice than those in PBS-treated mice, as determined by immunohistochemical (IHC) staining. Furthermore, enhanced immune complex deposition and citH3 expression can also be identified in pristane-injected mice when compared with their PBS-treated counterparts, as shown by IHC staining and immunoblotting (Figure 1D,E). We also found enhanced accumulation of neutrophils in pristane-injected WT mice when compared with their PBS-treated counterparts, as shown by immunofluorescent staining (Figure 1F, left and middle, white and red boxes). Interestingly, the galectin-3 stainings were enriched in the immune complex-positive areas and enhanced release of myeloperoxidase (MPO) in the haemorrhagic lung from pristane-treated WT mice (Figure 1D, anti-mouse IgG; Figure 1F, middle, red boxed areas), leading to a hypothesis that galectin-3 might be associated with the deposition of immune complex and NETs in the haemorrhagic lung of lupus mice. However, milder neutrophil infiltration and release of MPO were observed in lung tissues from pristane-treated Gal-3 KO mice as compared with pristane-treated WT mice by immunofluorescent analysis (Figure 1F, right, red boxed areas).

### 2.2. Amelioration of Murine Lupus Signs in the Absence of Galectin-3

The WT and Gal-3 KO mice were induced to evaluate their disease manifestations by pristane. Histopathologic analysis of lung tissues from Gal-3 KO mice revealed milder DAH and inflammation compared with those from WT mice. (Figure 2A). More WT mice died around week 4 after pristane treatment, but not in Gal-3 KO mice (Figure 2B). Furthermore, Gal-3 KO mice exhibited a trend toward reduced glomerular injury seen histologically and milder proteinuria than WT mice (Figure 2C,D). Levels of circulating anti-RNP antibody were lower in galectin-3 KO mice than their WT counterparts (Figure 2E). Interestingly, we found the expression of citH3 was decreased in lung tissue extracts from Gal-3 KO mice when compared with WT mice (Figure 2F), suggesting that galectin-3 might be associated with NETosis and exacerbate disease signs in lupus mice.

### 2.3. Galectin-3 Mediates NET Formation in Neutrophils from Human and Mice

To access the role of galectin-3 in NETosis, we treated neutrophils from healthy individuals with PMA for 4 h and identified the cellular localization of galectin-3 and MPO. Interestingly, we found that galectin-3 and MPO resided in NETs while neutrophils underwent NETosis (Figure 3A). Furthermore, increased expression of galectin-3 can also be observed in PMA-stimulated neutrophils (Figure 3A). Compared with glucose, NETosis was reduced in response to lactose treatment (Figure 3B) and was absent in neutrophils from Gal-3 KO mice treated with LPS compared with those from WT mice (Figure 3C). Reduced citH3 expression was also observed in the above experimental settings (Figure 3D, 17 kDa). Furthermore, the externalized galectin-3 signals in NETs can be co-localized with the signals from the sera of patients with SLE when compared with those from normal donors (Figure 3E).

## 3. Discussion

Emerging pieces of evidence have highlighted galectin-3 as an associate factor of SLE manifestations. Patients with LN have higher renal galectin-3 expression scores than normal donors, and the scores are correlated with anti-dsDNA titers and complement 3 and 4 levels [15]. The galectin-3 positive cells are mainly located in capillary loops, in the mesangium, on the parietal side of Bowman’s capsules, and in the crescents [15]. Furthermore, LN patients also have higher serum levels of galectin-3 and anti-galectin-3 antibody than healthy controls [15,16]. Our study showed that the expression levels of galectin-3 were higher in PBMCs of SLE patients than in normal controls, and the levels were positively correlated with SLEDAI-2k (Figure 1A). Anti-galectin-3 antibody levels were also higher in the sera of SLE patients than in normal controls (Figure 1C). Shi and colleagues found that the immune response mediated by anti-galectin-3 antibody plays a key role in the pathogenesis of SLE skin lesions [17]. Interestingly, galectin-1 and anti-galectin-1 antibody levels were significantly higher in SLE patients than in healthy individuals [24]; however, there is no correlation of the two molecules with disease activity [24]. These clinical findings revealed critical roles of galectin-3 to the SLE disease phenotypes.

DAH is a life-threatening complication of SLE with a low frequency, ranging from 1 to more than 5.4%, but a relatively high mortality rate from 23–92% [25]. It is suggested that enhanced infiltration of leukocytes into the lungs occurs, with immune complex deposition and capillarity in some lupus patients. However, the mechanism underlying molecular levels remains unclear. We found the galectin-3 stainings were enriched in the immune complex-positive areas of the haemorrhagic lungs from pristane-induced lupus mice (Figure 1D). The immunofluorescence double-staining was performed in the NETotic cells incubated with the indicated donors’ sera and anti-galectin-3 as primary antibodies; we observed co-localization of galectin-3 and anti-galectin-3 antibody residing in NETs (Figure 3E, merge). These data suggested that galectin-3 might contribute to enhanced immune complex deposition in NETs from haemorrhagic lungs of SLE. In addition to apoptotic or necrotic cell debris [9,10], NETs also provide the source of autoantigens and are supposed to be recognized by autoantibodies against neutrophil proteins such as lactoferrin [26], myeloperoxidase (MPO) [27], and elastase [28], which are known as anti-neutrophil cytoplasmic antibodies (ANCAs) [29]. Neutrophil-releasing NETs have been found in renal biopsies from patients with ANCA-positive vasculitis, whereas these NETs are enriched in MPO [30]. In supporting this process from our findings, we suggested that neutrophil galectin-3 might be an autoantigen that could be recognized by the anti-galectin-3 autoantibody, leading to immune complex deposition and organ damage. 

More importantly, intriguing evidence has emerged to reveal that neutrophil-specific genes are highly expressed in PBMCs from patients with SLE because a unique cell type designated low-density granulocytes (LDGs) can be found in mononuclear cell fractions. LDGs are primed to form exuberant NETs in the absence of stimulation and display enhanced ability to produce proinflammatory cytokines and matrix metalloproteinases that are highly toxic to endothelial cells [11,31]. Therefore, the possibility exists that this neutrophil subset has an important role in the induction of cardiovascular damage in patients with SLE by promoting endothelial damage and inflammation. This is supported by the finding that high numbers of LDGs correlate with vascular inflammation in SLE patients [31]. Furthermore, LDGs externalize enhanced levels of autoantigens, complement proteins, granular antimicrobial peptides such as MPO, neutrophil elastase, and matrix metalloproteinases within NETs [12,32,33,34]. We showed spontaneous NETosis in PBMCs of patients with SLE, but not in normal donors (Figure 3E, merge). Interestingly, we also found that galectin-3 resided in NETs while neutrophils underwent NETosis (Figure 3A), and the expression levels of galectin-3 were higher in PBMCs of SLE patients than those in normal donors (Figure 1A). Taken together, we suggested that galectin-3 might contribute to enhanced levels of immune complex deposition in NETs from LDGs of patients with SLE. Accordingly, further purification of LDGs from lupus patients should be applied to confirm this hypothesis. 

We found that NETosis could be reduced in neutrophils from Gal-3 KO mice and in lactose-treated human neutrophils compared with their control counterparts (Figure 3B,C), which suggested that galectin-3 activity might be responsible for the development of NETosis. Furthermore, citH3 expression levels were reduced in lung tissue extracts (Figure 2F) and neutrophils (Figure 3D) from Gal-3 KO mice when compared with WT mice after stimulation. The conversion of arginine residues to citrulline in core histones by protein arginine deiminase 4 (PAD4) is necessary for decondensation of chromatin before releasing as NETs [7]. A more recent finding indicates that galectin-3 and PAD4 may be involved in the pathogenesis of periodontal disease due to their elevated levels in periodontal disease [35]. Furthermore, galectin-3 induces ROS production by inflammatory peritoneal neutrophils from *Toxoplasma gondii*-infected mice [20]. Whether the two molecules could interact with each other for developing NETs is worth exploring, and the underlying mechanisms should also be explored in the future. 

This study contained limitations. First, Figure 3D shows the externalized galectin-3 signals in NETs can be co-localized with the signals from the sera of patients with SLE; the limitation of this data is the possibility that the signals of anti-dsDNA antibodies or anti-NETs antibodies can be co-localized with the galectin-3 signals. Although we found that galectin-3 resided in NETs while neutrophils underwent NETosis (Figure 3A), further experiments would be required to prove galectin-3 as an autoantigen residing in NETs, such as isolating galectin-3 from NETs and then performing anti-galectin-3 Ab ELISA as depicted in Figure 1C. Second, a contradictory scenario in Gal-3 KO lupus mice has been reported by Beccaria and colleagues [36]. They discovered that aged Gal-3 KO mice (8-month-old) developed spontaneous autoimmunity, including the presence of antinuclear antibodies and kidney damage. Compared with our animal model, pristane-induced Gal-3 KO mice (8-week-old) had rescued SLE-associated DAH, which occurred very early in the disease course (Figure 2A,B). According to our previous analysis for larger-scale series of SLE-associated DAH patients in the recent decade [37], the average age is around 13–38 years old, and the sex is female-dominant. This may explain SLEDAH usually occurring in young female patients, which could match our clinical and animal findings. Furthermore, target cells are different. In pristane-induced DAH, NETosis and NETs are dominant in the lung of lupus mice [38,39], and Gal-3 KO mice induced by pristane have diminished neutrophil numbers in the peritoneal cavities [19]. However, aged Gal-3 KO mice develop a lupus-like disease through increased percentages of germinal center B cells, antibody secreting cells, and higher concentrations of immunoglobulins and IFN-γ in serum compared with their WT counterparts [36]. Taken together, we suggested that galectin-3 might have a biphasic role in SLE, which contributes to DAH and NETosis in young female adults and mice in the early disease stage. Third, the NET detection approach can be improved. Although we showed extruded DAPI and MPO signals in NETs while neutrophils underwent NETosis in vitro (Figure 3A), and enhanced release of MPO in the haemorrhagic lung from pristane-treated WT mice in vivo (Figure 1F), quantitative fluorescence intravital lung microscopy (qFILM) can be used to observe real-time NETs in the lungs of lived mice by in vivo staining of neutrophil elastase, citrullinated histones, and neutrophils, as the state-of-the-art paper published by Vats and colleagues [40]. 

## 4. Materials and Methods

### 4.1. Patients and Normal Controls

Twenty-six patients fulfilling the American College of Rheumatology revised criteria for SLE, 23 females and 3 males aged from 19 to 62 years (38.2 ± 9.6), were enrolled into this study. Another 26 age- and sex-matched healthy individuals without known medical illnesses and not taking any medications, served as normal controls in this study. Clinical activities of patients were assessed by SLEDAI-2 K [41], and their activity scores were from 1 to 13 (7.0 ± 0.7). Thirteen patients had renal involvement, and their diagnosis of LN was according to the histopathological findings of renal biopsy. Another 13 cases had no abnormalities in renal profile including urine analyses and blood examinations. There was no observed DAH manifestation in all of the enrolled patients. In addition, LN patients had higher SLEDAI-2 K scores than those without renal involvement (9.5 ± 0.6 versus 4.5 ± 0.7, *p* < 0.001). 

### 4.2. Ethic Statement

The human study complied with the Declaration of Helsinki. The Institutional Review Board of National Cheng Kung University Hospital approved the research protocol for collection of venous blood samples with written informed consent obtained from the study subjects. The following animal experiments were done strictly in accordance with protocols approved by the Institutional Animal Care and Use Committee of National Cheng Kung University. 

### 4.3. Detection of Galectin-3 and Anti-Galectin-3 Antibody Levels in Patients with SLE

Blood samples from SLE patients and normal donors were collected and their PBMCs were isolated by Ficoll-Paque PLUS (ThermoFisher Scientific, Waltham, MA, USA). Total RNA was isolated from PBMCs with TRIzol reagent (Cat.#15596026, ThermoFisher Scientific) and complementary DNA was synthesized using a Verso cDNA synthesis kit (Cat.#AB1453A, ThermoFisher Scientific), and qRT-PCR was performed using Fast SYBR Green Master Mix (Cat.#4385612, ThermoFisher Scientific) to quantitate human galectin-3 gene expression levels with primer pairs 5′-GGCCACTGATTGTGCCTTAT-3′ and 5′-GAAGCGTGGGTTAAAGTGGA-3′ and GAPDH was used as an internal control. The comparative *Ct* method was used to calculate the relative abundance of each gene compared with GAPDH expression. Serum samples from SLE patients and normal donors were subjected to anti-galectin-3 antibody level assay. Briefly, the 96-well microplate was coated with 100 ng human galectin-3 proteins overnight, blocked with 1% bovine serum albumin for 1 h, and added with diluted serum samples from normal donors and SLE patients for 2 h. After incubation, the plate was washed with phosphate-buffered saline (PBS) solution containing 1% tween 20, and added with anti-human peroxidase antibodies (Cat.#109-035-088, Jackson ImmunoResearch Laboratories, West Grove, PA, USA) for 2 h. Tetramethylbenzidine as a chromogen substrate was finally added to the plate in which the reaction was stopped by providing 2N H_2_SO_4_ and read at absorbance 450 nm, with the correction wavelength set at 570 nm. 

### 4.4. Murine Pristane-Induced Lupus Model

Eight-week-old female Gal-3 KO [23] and WT C57BL/6 mice were intra-peritoneally injected with 500 μL pristane (Cat.#P2870, Sigma-Aldrich, St. Louis, MO, USA) or PBS once. WT mice died from DAH around week 4 after pristane injection were subjected to hematoxylin and eosin (H&E) staining. The other mice were sacrificed 64 days after pristane injection. The renal and lung tissue sections and extracts were subjected to H&E or IHC staining and immunoblotting. Proteinuria and anti-RNP antibody levels were determined by Pierce™ BCA Protein Assay Kit (Cat.#P2870, ThermoFisher Scientific) and anti-RNP antibody ELISA kit (Cat.#MBS006766, MyBioSource, San Diego, CA, USA).

### 4.5. H&E and IHC Stainings and Immunoblotting

After sacrificed, lung and renal tissues were isolated and fixed in 4% paraformaldehyde for 16–24 h, dehydrated, paraffin embedded, and longitudinally sectioned. Sequential 4-μM sections were stained with hematoxylin and eosin (H&E) and examined under a light microscope. For IHC staining, lung tissue sections from human and mice were deparaffinized in xylene, dehydrated in alcohol, epitope-unmasking by heating, immersed in H_2_O_2_, and stained with antibodies against galectin-3 (Cat.#H160, Santa Cruz Biotechnology, Dallas, TX, USA) and mouse IgG (Cat.#115-035-003, Jackson ImmunoResearch Laboratories). Cell lysates of neutrophils and lung tissue extracts were subjected to immunoblotting with antibodies against galectin-3 (Cat.#H160, Santa Cruz Biotechnology), citH3 (Cat.#ab5103, Abcam, Cambridge, UK), histone 3 (H3, Cat.#sc-10809, Santa Cruz Biotechnology), and quantitative control anti-β-actin-peroxidase antibodies (Cat.#A3854, Sigma-Aldrich). Protein–protein complexes were visualized with an ECL Plus System (Cat.#NEL105001EA, PerkinElmer, Waltham, MA, USA) and analyzed with a BioSpectrum Imaging System, UVP, for chemiluminescence detection. The semi-quantitative analyses of immunoblots were performed by Image J software (National Institute of Health, Bethesda, MD, USA), as described previously [12].

### 4.6. NETosis and Immunofluorescence Analyses

Normal human neutrophils were isolated by polymorphprep (Cat.#AXS-1114683, AXIS-SHIELD, Luna Place, Scotland) and confirmed by their polymorphonuclear phenotypes without monocyte contamination [12] and stimulated with phorbol myristate acetate (PMA, 100 μM, Cat.#p1585, Sigma-Aldrich) for 4 h (T4) to induce NETosis or analyzed at baseline (T0). NETotic cells were fixed with 4% paraformaldehyde and stained with DAPI for 10 min or with anti-human galectin-3 or anti-human MPO (Cat.#ab208670, Abcam) antibody and followed by Texas-red conjugated secondary antibody (Cat.#112-075-003, Jackson ImmunoResearch Laboratories) for immunofluorescence analysis. Neutrophils isolated from normal donors were treated with lactose or glucose (20 and 50 mM) at T4 and then fixed and stained with DAPI for 10 min. NETotic cells were quantitated by ImageJ system. Normal murine neutrophils were collected by intra-peritoneal injection of 3% thioglycolate (Cat.#108191, Merck, Darmstadt, Germany) into WT C57BL/6 mice for 24 h, and then their peritoneal cavities were washed by injecting 5 mL PBS. The fluids were subjected to polymorphprep for isolating neutrophils as described above. Murine neutrophils were stimulated with 20 μM of lipopolysaccharide (LPS, Cat.#L2008, Sigma-Aldrich) for 8 h to induce NETosis and observed as described above. Spontaneous NETosis was observed by isolating PBMCs from SLE patients or normal donors and analyzed at T0. Cells were stained with DAPI for 10 min or with rat anti-human galectin-3 antibody or serum from indicated donors and followed by Texas-red conjugated anti-rat or FITC-conjugated anti-human IgG secondary antibody (Cat.#ab97224, Abcam) for immunofluorescence analysis. For the stainings in lung tissue sections, preparations as described above were subjected to anti-mouse MPO (Cat.#ab208670, Abcam) and FITC conjugated anti-mouse Ly-6G (Cat.#RB68C5, ThermoFisher Scientific) antibodies, followed by the Texas red-conjugated secondary antibody (Cat.#111-075-003, Jackson ImmunoResearch Laboratories). Autofluorescence was quenched by treating each sections with the TrueBlack^®^ Lipofuscin Autofluorescence Quencher (Cat.#92401, Cell Signaling Technology, Danvers, MA, USA).

### 4.7. Statistical Analysis

Data are expressed as the mean ± SEM. Differences between patients and normal controls or between different patient groups were analyzed by the Mann-Whitney U test. Differences between two groups and among groups were analyzed using Student’s *t* test and one-way ANOVA followed by Dunnet multiple comparison tests, respectively (Prism 5.0). The significance of correlation between the expression levels of galectin-3 and SLE disease activity index 2000 (SLEDAI-2K) was calculated using Pearson’s correlation coefficient. The percent survival between WT and Gal-3 KO mice was compared by the log-rank (Mantel-Cox) test (Prism 5.0); *p* values less than 0.05 were considered significant.

## 5. Conclusions

In conclusion, by analyzing PBMCs and neutrophils from SLE patients and lupus mice, we found that galectin-3 mediates NETosis and acts as an autoantigen within NETs. These findings may contribute to developing novel pharmacological therapies targeting galectin-3-mediated NETosis for SLE therapy.

## Figures and Tables

**Figure 1 ijms-24-09493-f001:**
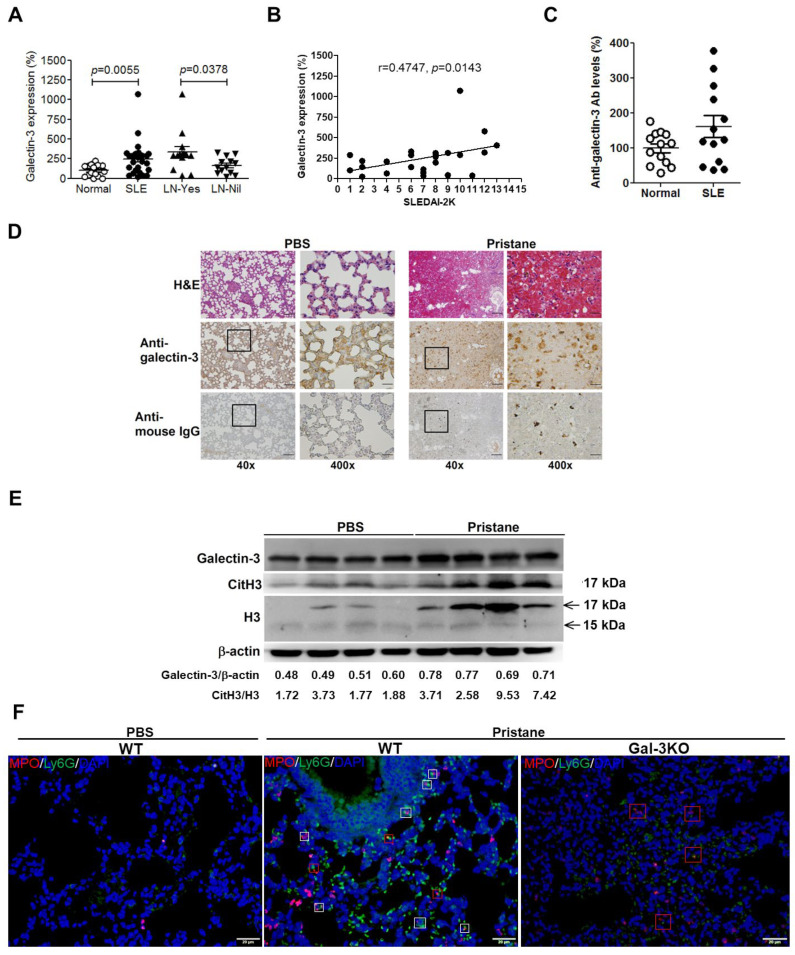
Expression of galectin-3 in patients with systemic lupus erythematosus (SLE) and mice with pristane-induced lupus. (**A**) Quantitative reverse transcription-polymerase chain reaction (qRT-PCR) of galectin-3 in PBMCs of SLE patients (SLE) and in the same cohort with (LN-Yes) or without (LN-Nil) lupus nephritis and normal donors (Normal). (**B**) The significance of correlation between the expression levels of galectin-3 and SLE disease activity index 2000 (SLEDAI-2K) was calculated using Pearson’s correlation coefficient. (**C**) Anti-galectin-3 antibody levels in sera of SLE patients and normal donors. (**D**) Female mice aged 8 weeks were injected with 500 μL of pristane or phosphate-buffered saline (PBS) via the intra-peritoneal routes. Mice died from diffuse alveolar haemorrhage (DAH) around week 4 after pristane injection, as demonstrated by H&E staining. Immunohistochemical stainings of galectin-3 and moue IgG Fc in lung tissue sections from PBS and pristane-treated mice. Boxed areas are shown at higher magnification in the panels next to them. (**E**) Expression of galectin-3, citrullinated histone 3 (citH3), and histone 3 (H3) in lung tissue extracts from PBS and pristane-treated mice (n = 4). (**F**) Immunofluorescent stainings of myeloperoxidase (MPO, Texas red), and Ly6G (FITC) in lung tissue sections from PBS and pristane-treated WT and galectin-3 knockout (Gal-3 KO) mice. Boxed areas indicate MPO and Ly6G double-positive signals (×400 magnification). DAPI indicates nuclear staining. Values were presented as mean and SEM. Bars shown on the photomicrographs at ×40, ×100, and ×400 magnifications correspond to 500, 200, and 50 μm, respectively.

**Figure 2 ijms-24-09493-f002:**
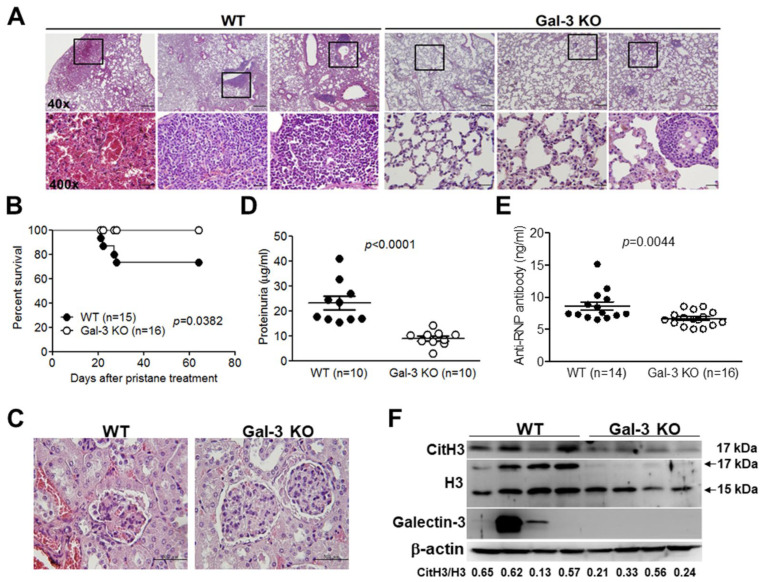
Associations of galectin-3 and lupus manifestations in pristane-treated galectin-3 knockout (Gal-3 KO) mice. Female wild-type (WT) and Gal-3 KO mice aged 8 weeks were injected with 500 μL of pristane via the intra-peritoneal routes. (**A**) Representative images of DAH and inflammation from pristane-treated Gal-3 KO mice compared with WT counterparts, as demonstrated by H&E staining (n = 3). Boxed areas are shown at higher magnification in the panels beneath them (**B**) Percent survival in pristane-induced WT and Gal-3 KO mice. (**C**) Representative images of renal sections from pristane-treated Gal-3 KO mice compared with their WT counterparts, as demonstrated by H&E staining (×400 magnification). (**D**) Levels of proteinuria in pristane-induced WT and Gal-3 KO mice. (**E**) Levels of anti-RNP antibodies in sera from pristane-induced WT and Gal-3 KO mice, as determined by ELISA. (**F**) Expression of galectin-3, citH3, and H3 in lung tissue extracts from pristane-induced WT and Gal-3 KO mice, as determined by immunoblotting (n = 4). Values were presented as mean and SEM. Bars shown on the photomicrographs at ×40, ×100, and ×400 magnifications correspond to 500, 200, and 50 μm, respectively.

**Figure 3 ijms-24-09493-f003:**
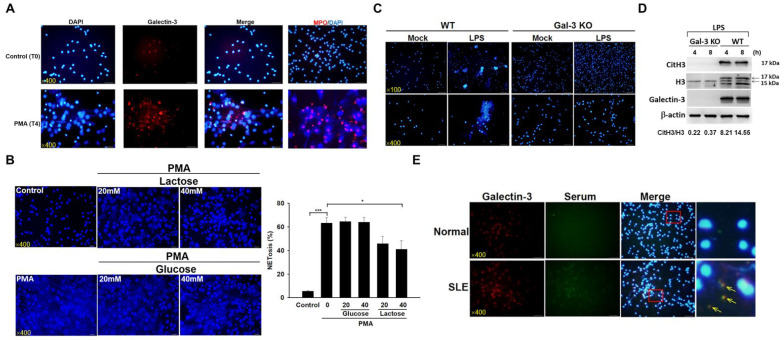
Effects of galectin-3 on NETosis. (**A**) Representative images of neutrophils isolated from normal donors and analyzed at baseline (T0) or after stimulation with phorbol myristate acetate (PMA, 100 μM) for 4 h (T4). Cells were stained with DAPI for 10 min or with anti-human galectin-3 or with anti-human MPO antibody and followed by Texas-red conjugated secondary antibodies for immunofluorescence analysis. (**B**) Representative images and quantification of neutrophils isolated from normal donors and treated with lactose or glucose (20 and 50 mM) at T4. Cells were stained with DAPI for 10 min. Values were presented as mean and SEM (n = 3). (**C**) NETosis and (**D**) Galectin-3, citH3, and H3 expression in lipopolysaccharide (LPS, 20 μM)-treated neutrophils from WT and Gal-3 KO mice. (**E**) Representative images of PBMCs isolated from SLE patients or normal donors and analyzed at T0. Cells were stained with DAPI for 10 min or with rat anti-human galectin-3 antibody or sera from indicated donors and followed by Texas-red conjugated anti-rat or FITC-conjugated anti-human IgG secondary antibody for immunofluorescence analysis. Higher-magnification views of the merged images are shown on the far right. The yellow arrows indicate immune complex deposition. Bars shown on the photomicrographs at ×100 and ×400 magnifications correspond to 200 and 50 μm, respectively. * *p* < 0.05, *** *p* < 0.001.

## Data Availability

The data that support the findings of this study are available on request from the corresponding author.

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
