# Peer review of "Galectin-3 Mediates NETosis and Acts as an Autoantigen in Systemic Lupus Erythematosus-Associated Diffuse Alveolar Haemorrhage"

_ijms, 2023, doi:10.3390/ijms24119493_

Round 1
Reviewer 1 Report (Previous Reviewer 2)
The Authors performed additional staining for MPO along with Ly6G as well as changed some sentences across the manuscript to make it of better shape. The paper still has limitations, but I am leaning toward acceptance of it.
Before that please correct some minors:
1) Page 10, line 30 - approach of isolation of galectin-3 from NETs is very controversial since NETs are web-like structures mostly chromatin, exDNA and some key proteins. I do not think that gal-3 is there at all. Please rephrase.
2) Please create a paragraph that will sum up all the study limitations. In this paragraph please describe the weakness of the NETs detection approach and please refer the readers to a state-of-the-art paper describing multiple approaches for NETs detection: https://doi.org/10.1182/blood.2021014552
Best.
Author Response
Comments and Suggestions for Authors
The Authors performed additional staining for MPO along with Ly6G as well as changed some sentences across the manuscript to make it of better shape. The paper still has limitations, but I am leaning toward acceptance of it.
Before that please correct some minors:
Page 10, line 30 - approach of isolation of galectin-3 from NETs is very controversial since NETs are web-like structures mostly chromatin, exDNA and some key proteins. I do not think that gal-3 is there at all. Please rephrase.
Response: Thank you very much for raising these critical issues. We moved the sentence to limitation (Page 11, line 16) and rephrased it. In fact, we began this study when we found that galectin-3 resided in NETs while neutrophils underwent NETosis (Fig. 3A). We are very excited about this finding and the data has ever been discussed with Prof. Fu-Tong Liu, who’s the expert in the galectin field. He said that similar phenomena had been observed by him. As mentioned by reviewer 1 that “NETs are web-like structures mostly chromatin, exDNA and some key proteins”. We also included a similar sentence” NETs also provide the source of autoantigens and are supposed to be recognized by autoantibodies against the neutrophil proteins like lactoferrin[27], myeloperoxidase (MPO)[28], and elastase[29], which are known as anti-neutrophil cytoplasmic antibodies (ANCAs)[30]” (page 10, line 30-33). On page 10, line 35, “In supporting this process from our findings, we suggested that neutrophil galectin-3 might be an autoantigen that could be recognized by the anti-galectin-3 autoantibody, leading to immune complex deposition and organ damage.” Galectin-3 has been recognized as an autoantigen in SLE skin lesions (ref. 17), and in this study, we proposed that galectin-3 would reside in NETs and act as an autoantigen. We know that there have been a lot to do according to the present findings. Therefore, we will try to excise NETs from netting neutrophils, which can be achieved by treating DNAse1 (J Vis Exp. 2015 Apr 16;(98):52687) and find out if galectin-3 resides in it.
Please create a paragraph that will sum up all the study limitations. In this paragraph please describe the weakness of the NETs detection approach and please refer the readers to a state-of-the-art paper describing multiple approaches for NETs detection: https://doi.org/10.1182/blood.2021014552
Response: Thank you very much for providing such a state-of-the-art paper describing multiple approaches for NETs detection, we have learned a lot and will try to improve our techniques. The limitations of this study, including the weakness of the NETs detection approaches have been included on page 11, lines 16-45.
Thanks again for reviewing our manuscript.

Reviewer 2 Report (Previous Reviewer 1)
The authors addressed all my concerns in the resubmitted version. I have no further questions.
Author Response
Thank you very much for taking the time to review this manuscript for us. We appreciate your kind help.
This manuscript is a resubmission of an earlier submission. The following is a list of the peer review reports and author responses from that submission.
Round 1
Reviewer 1 Report
In this paper, the authors reported that Galectin-3 levels are higher in SLE patients compared with controls and positively correlated with disease activity. Gal-3 KO mice have higher percent survival and lower DAH, LN proteinuria, and anti-RNP antibody levels than WT mice induced by pristane. Furthermore, galectin-3 resides in NETs while human neutrophils undergo NETosis. All these data added our knowledge regarding NETosis in SLE. Here are my concerns/ comments below.
1. Please provide the information regarding the demographic and clinical characteristics of SLE patients.
2. Figure 1C: The anti-galectin-3 antibody levels were variable in SLE patients. Were these antibodies levels associated with SLEDAI, anti-dsDNA, anti-RNP antibodies or C3/C4 levels? Were there any differences regarding other clinical information between the patients with higher or lower anti-galectin-3 antibody levels?
3. Figure 2E: The authors found pristane treated Gal-3 KO mice produced lower levels of anti-RNP antibodies. It has been reported that TLR-7 is specifically required for the production of RNA-reactive autoantibodies and the development of glomerulonephritis in pristane-induced murine lupus (Savarese E. Arthritis & Rheumatology. 2008; Wang T. Frontiers in immunology. 2019). Do the authors believe that there is interaction between galectin-3 and TLR7 in the pathogenesis of SLE?
4. Figure 3E: how did the authors determine the antibodies bound to NETs were anti-galectin-3 antibodies? Is there possibility that they are anti-dsDNA antibodies or anti-NETs antibodies (Zuo Y. Arthritis & Rheumatology. 2020)?
Author Response
Comments and Suggestions for Authors
In this paper, the authors reported that Galectin-3 levels are higher in SLE patients compared with controls and positively correlated with disease activity. Gal-3 KO mice have higher percent survival and lower DAH, LN proteinuria, and anti-RNP antibody levels than WT mice induced by pristane. Furthermore, galectin-3 resides in NETs while human neutrophils undergo NETosis. All these data added our knowledge regarding NETosis in SLE. Here are my concerns/ comments below.
Response: Thank you very much for taking your precious time to review our manuscript, we will respond to your valuable suggestions as follows,
- Please provide the information regarding the demographic and clinical characteristics of SLE patients.
Response: Please check section 4.1. Patients and normal controls on page 8, lines 17-27, we have provided the information of patients’ genders, ages, renal involvement or not, no observed DAH manifestation, and SLEDAI-2K scores.
- Figure 1C: The anti-galectin-3 antibody levels were variable in SLE patients. Were these antibodies levels associated with SLEDAI, anti-dsDNA, anti-RNP antibodies or C3/C4 levels? Were there any differences regarding other clinical information between the patients with higher or lower anti-galectin-3 antibody levels?
Response: Thank you very much for raising these points. In fact, our clinical findings shown in Figures 1a-1c have been previously confirmed by others (references 15, 17, and 25), which were also discussed in our manuscript (page 6, lines 20-34). Anti-galectin-3 antibody plays a key role in the pathogenesis of SLE skin lesions (Page 6, lines 29-31 and ref.17). Furthermore, patients with LN have higher renal galectin-3 expression scores than normal donors, and the scores are correlated with anti-dsDNA titers, complement 3 and 4 levels (Page 6, line 21-23 and ref.15). Therefore, we do not focus too much on our clinical findings in which the critical concepts have been almost proven before.
- Figure 2E: The authors found pristane treated Gal-3 KO mice produced lower levels of anti-RNP antibodies. It has been reported that TLR-7 is specifically required for the production of RNA-reactive autoantibodies and the development of glomerulonephritis in pristane-induced murine lupus (Savarese E. Arthritis & Rheumatology. 2008; Wang T. Frontiers in immunology. 2019). Do the authors believe that there is interaction between galectin-3 and TLR7 in the pathogenesis of SLE?
Response: Thank you very much for providing the valuable information. After surveying pieces of literature, we found that galectin-3 can bind TLR4 and act as an alarmin in traumatic brain injury (Sci Rep. 2017;7:41689) and mediates TLR4/NF-ĸB signaling (BMC Cancer. 2018;18:580). Since TLR7 is an endosomal innate immune sensor, it can access galectin-3 easier than cell membrane TLR like TLR4. To our knowledge, galectins are basically not secretory proteins that contain no signal peptide, intracellular galectins can bind their intracellular partners and control cell fate in a carbohydrate-dependent or independent manner, which is proposed by the expertise Prof. Fu-Tong Liu in the galectin field (Journal of Biomedical Science 28: 1-9). To this aspect, we plan to examine if galectin-3 can bind intracellular PAD4 to influence NETosis, as discussed on page 7, lines 40-52. The reviewer’s interesting points are worth expecting some proof in the future.
- Figure 3E: how did the authors determine the antibodies bound to NETs were anti-galectin-3 antibodies? Is there possibility that they are anti-dsDNA antibodies or anti-NETs antibodies (Zuo Y. Arthritis & Rheumatology. 2020)?
Response: Thank you very much for raising this critical point. Indeed, they might be anti-dsDNA antibodies or anti-NETs antibodies, and their signals co-localized with the galectin-3 signals. Therefore, we removed the description “indicating galectin-3 might be an autoantigen and that could be recognized by the anti-galectin-3 autoantibodies in sera from lupus patients” on page 5, lines 1-2 and modified the sentence on page 6, line 28-30. We further included the following limitation” Although the limitation of this data is the possibility that the signals of anti-dsDNA antibodies or anti-NETs antibodies can be colocalized with the galectin-3 signals, further experiments will be required, such as isolating galectin-3 from NETs and then performing anti-galectin-3 Ab ELISA as depicted in Figure 1C.” on page 7, lines 7-10.

Reviewer 2 Report
The presented paper focuses on the role of Galectin-3 in mediating NETosis under SLE conditions.
The discussed area is the emerging field of medicine, thus, every paper contributing to a better understanding of the NETosis process in SLE is significant and welcome to the field. I found the paper to a good merit and scientific soundness.
Albeit, I have some majors, that must be fixed during the peer review process.
1) The absence of citH3/H3 in neutrophils of Gal3 KO mice (line 39) suggests that the whole NEtosis process is halted. It is not possible due to the multi-faced pathways leading to this process. This statement must be fixed and followed.
2) How was the purity of isolated from human blood neutrophils checked? Without confirmation of the purity, there is a high risk of contamination by monocytes that can give you false-positive results.
3)Figure 3 might be very misleading. To determine the presence of netosis the authors must perform additional staining for more than 1 marker of netosis. Solely citH3 is not enough. I suggest adding NE/MPO and exDNA and then, observing the colocalization of them indicating the process called NETosis.
4) Fig 2F shows the westerns from the whole lung lysate. It is not appropriate for this paper. Westerns must be done on isolated neutrophils.
5) Fig 2A has no scale
6) Fig 1E - again, the whole lung tissue is not relevant to the studied phenomena. Neutrophils westerns must be performed
7) Basic characteristic of enrolled subjects (SLE) must be included.
Author Response
Comments and Suggestions for Authors
The presented paper focuses on the role of Galectin-3 in mediating NETosis under SLE conditions.
The discussed area is the emerging field of medicine, thus, every paper contributing to a better understanding of the NETosis process in SLE is significant and welcome to the field. I found the paper to a good merit and scientific soundness.
Albeit, I have some majors, that must be fixed during the peer review process.
Response: Thank you very much for taking your precious time to review our manuscript, we will respond to your valuable suggestions as follows,
- The absence of citH3/H3 in neutrophils of Gal3 KO mice (line 39) suggests that the whole NEtosis process is halted. It is not possible due to the multi-faced pathways leading to this process. This statement must be fixed and followed.
Response: Thank you very much for raising this critical point. We have modified the descriptions regarding this point throughout the whole manuscript. On page 1, line 39, “reduced”; page 5 line 27, “Reduced”; page 7, line 40, “ We found that NETosis could be reduced”.
- How was the purity of isolated from human blood neutrophils checked? Without confirmation of the purity, there is a high risk of contamination by monocytes that can give you false-positive results.
Response: Thank you for your critical concern. We have carefully isolated neutrophils by polymorphprep and their polymorphonuclear phenotypes were confirmed according to our previous publication (Ref. 12). We are sure there is no monocyte contamination in each experiment. We included the following sentence in section 4.6. NETosis and immunofluorescence analyses ” confirmed by their polymorphonuclear phenotypes without monocyte contamination [12]” on page 9, lines 33-34.
- Figure 3 might be very misleading. To determine the presence of netosis the authors must perform additional staining for more than 1 marker of netosis. Solely citH3 is not enough. I suggest adding NE/MPO and exDNA and then, observing the colocalization of them indicating the process called NETosis.
Response: Thank you for the suggestion. The conversion of arginine residues to citrulline in core histones by protein arginine deiminase 4 (PAD4) is necessary for the decondensation of chromatin before releasing as NETs, therefore, citH3 is a very specific marker for pristane-induced NETosis. We agree with the reviewer’s suggestion that additional staining for more than 1 marker of netosis may persuade readers about this phenomenon in this paper. However, according to our previous reports, DNA morphology and citH3 expression were enough to represent NETosis (Ref. 12).
4) Fig 2F shows the westerns from the whole lung lysate. It is not appropriate for this paper. Westerns must be done on isolated neutrophils.
6) Fig 1E - again, the whole lung tissue is not relevant to the studied phenomena. Neutrophils westerns must be performed.
Response: Thank you very much for raising the above 2 similar points. Our previous findings revealed citH3 expression levels were increased in lung tissue extracts of pristane-induced mice compared with control mice and distinct expression of citH3 colocalized with DNAs, favoring NETs formation, was identified in pristane-injected but not control mice (Ref. 12). In Figures 2F, and 3D, we demonstrated very specifically that citH3 expression levels were reduced in lung tissue extracts (Fig. 2F) and neutrophils (Fig. 3D) from Gal-3 KO mice when compared to WT mice after stimulation. NETosis is therefore reduced in neutrophils from Gal-3 KO mice after LPS stimulation (Fig. 3C). These data indicate that distinct citH3 expression may represent NETosis in vitro or in vivo, especially the role of galectin-3 involved in it.
5) Fig 2A has no scale
Response: Thank you for reminding us. We did show scale bars in Fig 2A and the information about the scale bars was shown on page 5, lines 17-19.
7) Basic characteristic of enrolled subjects (SLE) must be included.
Response: Please check section 4.1. Patients and normal controls on page 8, lines 17-27, we have provided the information of patients’ genders, ages, renal involvement or not, no observed DAH manifestation, and SLEDAI-2K scores.

Round 2
Reviewer 1 Report
The authors have addressed my concerns properly and made appropriate modifications to the manuscript. I have no further questions.
Reviewer 2 Report
This is an experimental paper, not the theoretical one.
the authors did not address any of my concerns by performing the required experiments.
I do not feel that the conclusions are supported by the presented results.
this paper must not be published until my majors are addressed by performing additional experiments as required.
Otherwise I will force to retract this paper if published